# Investigating the native functions of [NiFe]-CODH through genomic context analysis

**Maximilian Böhm, Henrik Land***

Molecular Biomimetics, Department of Chemistry – Ångström Laboratory, Uppsala University, Uppsala, Sweden

## eLife Assessment

This **valuable** work analyzes a large dataset of [NiFe]-CODHs, integrating genomic context, operon organization, and clade-specific gene neighborhoods to discern patterns of diversification and adaptation. A consistent examination of CODH genomic contexts, including CODH-HCP co-occurrence, informs interpretations of enzymatic activity, biotechnological potential, and differential functional roles, in line with current standards in genomic enzymology. With **solid** support, this work provides a broadly informative contribution to the field.

***For correspondence:**
henrik.land@kemi.uu.se

**Competing interest:** The authors declare that no competing interests exist.

## Abstract

Carbon monoxide dehydrogenases containing nickel-iron active sites ([NiFe]-CODHs) catalyze the reversible oxidation of CO to $CO_2$, representing key targets for biocatalytic $CO_2$ reduction. Despite dramatic differences in catalytic rates and $O_2$ tolerance between CODH variants, the molecular basis for this functional diversity remains poorly understood. We applied comparative genomics and synteny analysis to investigate the biochemical roles of CODH clades A-F using 1376 CODH and 1545 hybrid cluster protein sequences. Around 30% of genomes encode multiple CODH isoforms. Analysis revealed distinct gene clustering patterns correlating with biochemical function. Clades A, E, and F exhibit a degree of distributional exclusivity. Clades C and D frequently co-occur with active CODHs, suggesting auxiliary roles. Operon architecture analysis revealed functional specialization: clade A links to acetyl-CoA synthase; clades A, E, and F contain essential maturation machinery (CooC, CooJ, CooT) correlating with catalytic activity; clade B associates with transporters; clade C with electron transfer partners; clade D with transcriptional regulators. High CODH-HCP co-occurrence (except clade A) suggests functional or environmental interdependency. These findings establish clades A, E, and F as primary biocatalyst targets while defining regulatory functions for clades C and D, providing a genomics framework for predicting CODH phenotypes.

## Introduction

Genomic enzymology has been proven to help understand protein (super) families since the mid-1990s, helping to connect enzyme sequences to function through comparative genomics and neighborhood analysis (*Babbitt et al., 1996*; *Knox and Allen, 2023*). In this study, we are employing a genome neighborhood and co-occurrence analysis to help understand reactivity and functionality of the family of nickel-containing carbon monoxide dehydrogenases ([NiFe]-CODHs) and their relationship to hybrid cluster proteins (HCPs).

[NiFe]-CODHs are ancient and diverse enzymes that catalyze the interconversion between carbon monoxide (CO) and carbon dioxide ($CO_2$), a reaction of high interest for biotechnological applications, including $CO_2$ capture and conversion. Research on this enzyme spans over 60 years, and recent

studies have provided important biochemical insights, such as their turnover frequency, oxygen tolerance, and catalytic mechanism (*Basak et al., 2025*; *Can et al., 2014*). These properties vary greatly between enzymes, not only between separate phylogenetic clades but also within clades, making functional prediction from sequence alone challenging. Phylogenetic analyses of available [NiFe]-CODH (hereafter referred to as CODH) sequences with different focus such as gene transfer (*Techtmann et al., 2012*), primary structure (*Inoue et al., 2018*), biome distribution (*Inoue et al., 2022*), and human gut microbiome (*Katayama et al., 2024*) have enriched our understanding of this old and diverse enzyme family. From initially small data sets of 17 sequences (*Lindahl and Chang, 2001*) to datasets well above 5000 sequences (this study). It has been shown that up to eight distinct phylogenetic clades (*Figure 1*) can be distinguished with all of them having sequence variations while preserving the overall fold, as seen with cryo-electron microscopy (*Biester et al., 2024*) and X-ray crystallography (*Basak et al., 2025*; *Domnik et al., 2017*; *Gong et al., 2008*; *Jeoung et al., 2022*; *Jeoung and Dobbek, 2007*; *Wittenborn et al., 2020*).

The biochemical characterization of this enzyme family is still ongoing, and it shows a wide range of turnover frequencies, as well as different degrees of $O_2$ tolerance. For example, looking at two CODHs from *Carboxydothermus hydrogenoformans*: *Ch*CODH-II, a benchmark CODH known for its high CO oxidation activity but low $O_2$ tolerance, contrasts with *Ch*CODH-IV – another enzyme from the same

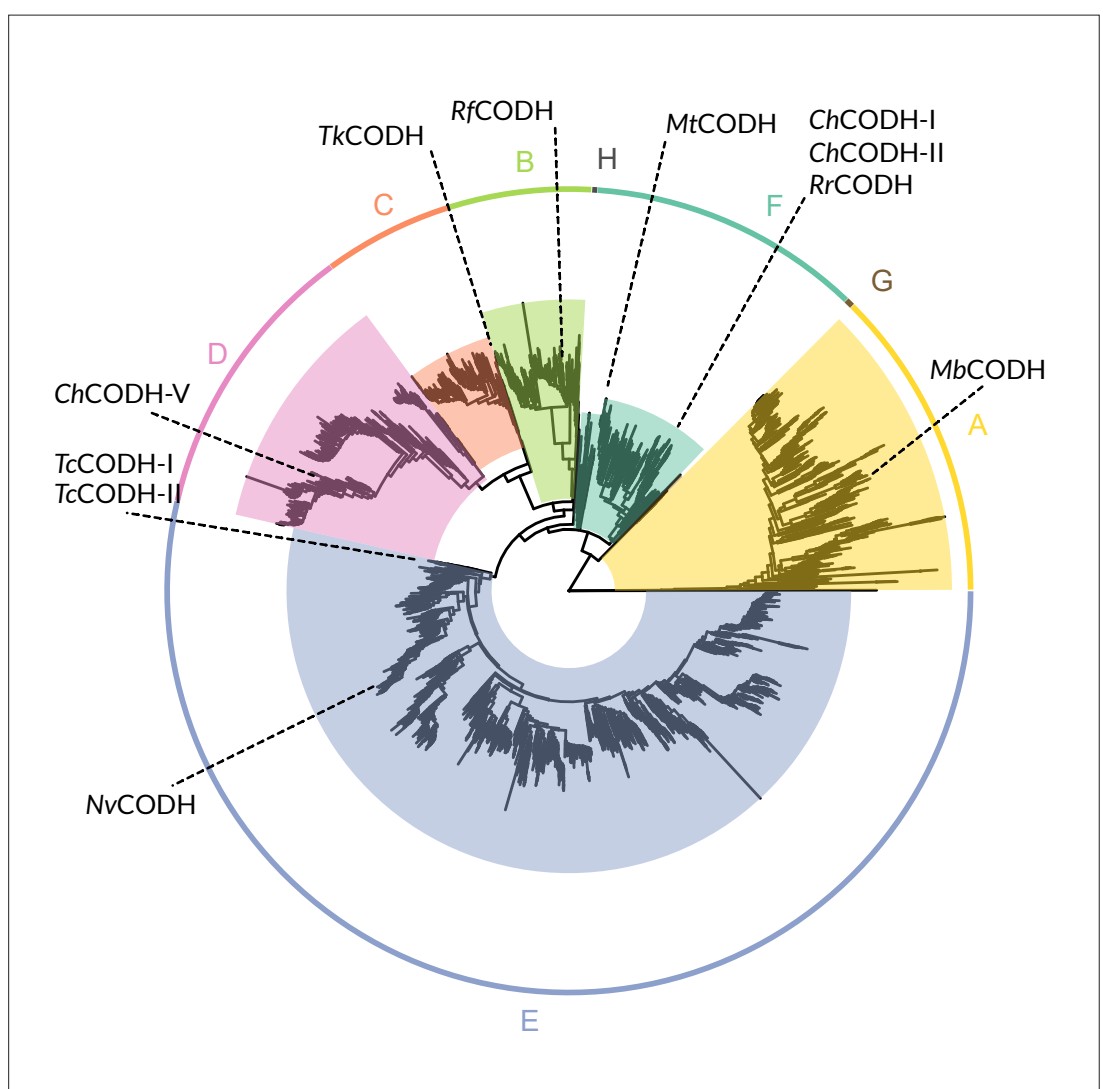

**Figure 1.** Schematic phylogenetic tree of [NiFe]-CODH, with selected CODHs marked with their respective positions. Tree was built using IQ-TREE, 1000 ultrafast bootstrap, containing 5508 putative CODH sequences and one outgroup (MBE6442607.1 hydroxylamine reductase [*Desulfovibrio desulfuricans*]) for rooting. A detailed searchable tree with bootstrap values can be found in *Supplementary file 5*.

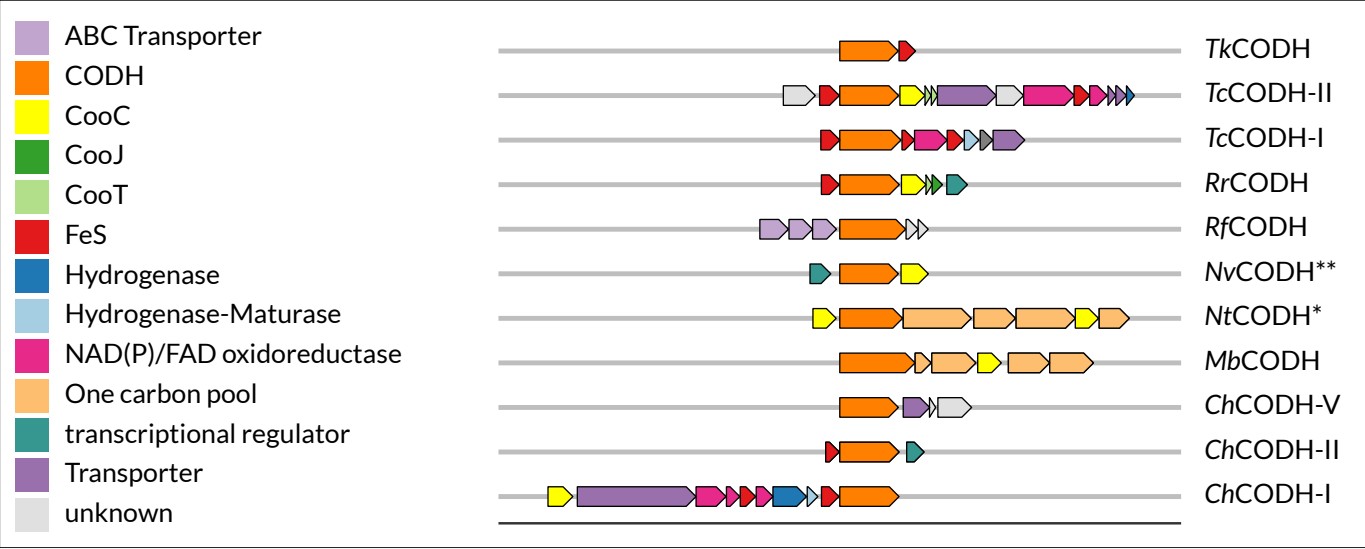

**Figure 2.** Operons of selected [NiFe]-CODH. * NtCODH, formally known as MtCODH, formerly known as CtCODH due to renaming of host organism (*Gtari and Ventura, 2025*). ** NvCODH, formerly known as DvCODH due to renaming of host organism (*Waite et al., 2020*).

clade and organism – which retains 20% of activity after 1 hr of $O_2$ exposure but displays reduced CO oxidation capacity with an increased activation barrier and much lower $K_M$ (*Domnik et al., 2017*), both belonging to clade F (*Figure 1*). Similarly, a less active CODH from clade E, *Nv*CODH (formerly known as *Dv*CODH) from *Nitratidesulfovibrio vulgaris*, has been reported to fully reactivate after initial inactivation by $O_2$ exposure (*Hadj-Saïd et al., 2015*). Also, the two CODHs from *Thermococcus* sp. AM4, *Tc*CODH-I, and *Tc*CODH-II belonging to clade E react slower with $O_2$ compared to *Ch*CODH-II, but have overall equal $O_2$ sensitivity (*Benvenuti et al., 2020*).

In addition to the previously mentioned diversity within CODH's clades or organisms, with regard to activity and oxygen tolerance, it is known that some CODHs rely on maturases for full activation, while others do not. For example, *Rr*CODH, from the phototroph *Rhodospirillum rubrum*, needs to be expressed together with three maturases (CooC, CooJ, and CooT) in order to be isolated in an active form (*Kerby et al., 1997*). A similar situation arises for *Nv*CODH; however, its genomic neighborhood (*Figure 2*) only contains one maturase (CooC) which is required for active production (*Hadj-Saïd et al., 2015*). On the contrary, *Ch*CODH-II can be heterologously expressed without co-expression of any maturases (*Merrouch et al., 2018*). Also, *Ch*CODH-I needs to be co-expressed with CooC in order to reach high activity, but it can also be expressed without it, albeit with reduced activity (*Inoue et al., 2014*). Interestingly, much of the diversity with regard to activity, $O_2$ tolerance, and maturase dependence does not only occur between the different clades but also within them.

Due to the homology between the CODHs in this study and the fact that active CODHs have been demonstrated from several of the clades, it is reasonable to assume that CODHs from all clades are able to interconvert $CO_2$/CO. However, a recent study by Dobbek and co-workers showed that *C. hydrogenoformans* CODH-V (*Ch*CODH-V) from clade D was not able to perform this reaction (*Jeoung et al., 2022*). They showed that this enzyme has a closer similarity to the family of HCPs, due to its morphing active site, composed of iron, sulfur, and oxygen, responding with structural and stoichiometric changes upon redox shifts. A connection between HCPs and CODHs has been pointed out previously by *Inoue et al., 2018*, due to their close phylogenetic relationship, and was further discussed by *Fujishiro and Takaoka, 2023*. HCPs can be divided into three phylogenetic classes, of which class III exhibits a homodimeric structure like CODH. Generally, the two enzyme families share a similar overall fold while their active sites differ greatly, both in terms of amino acid and metallocofactor composition. Similar to *Ch*CODH-V, HCPs do not catalyze $CO_2$/CO interconversion, but they do display a range of activities at low rate, such as hydroxylamine reductase, peroxidase, nitric oxide reductase, and S-nitrosylase activity. The main natural function of HCPs is debated, but it was recently established that it is most likely a nitric oxide reductase involved in nitric oxide detoxification (*Hagen, 2022*).

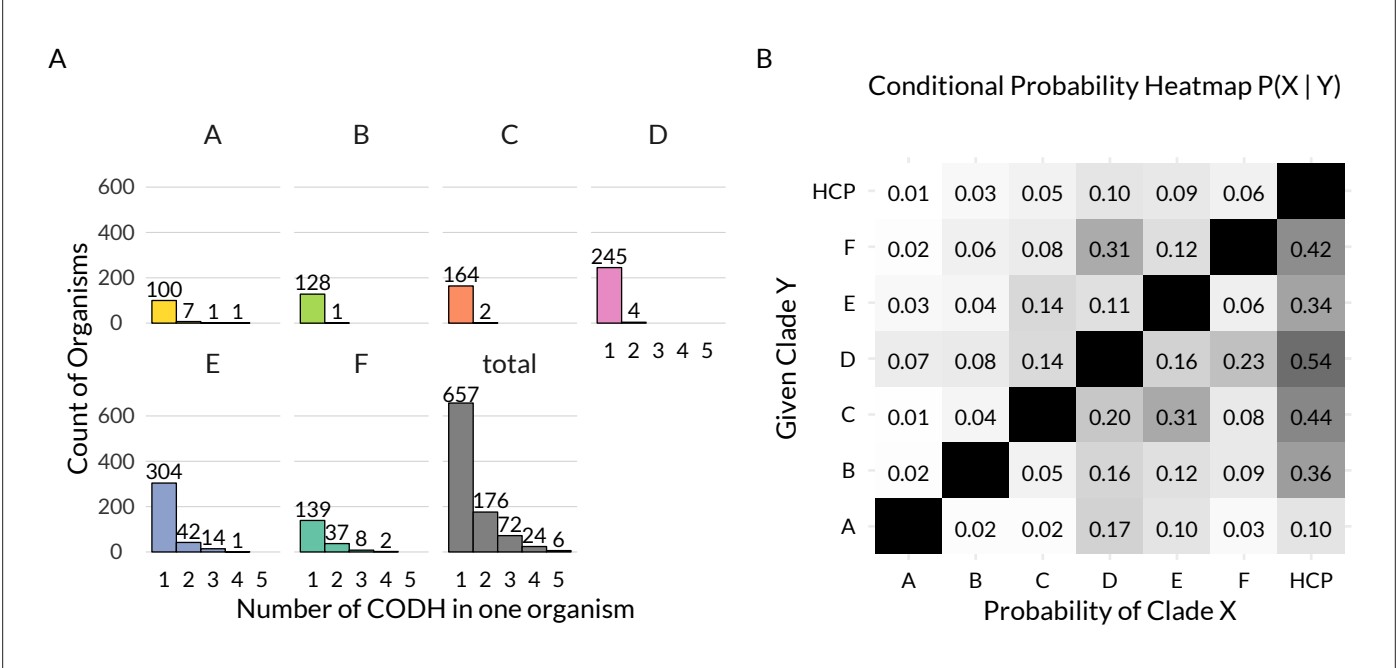

**Figure 3.** CODH co-occurence within organisms. (**A**) Frequency of X amounts of [NiFe]-CODH from the same clade co-occurring in the same organism. (**B**) Probability matrix of [NiFe]-CODH from different clades co-occurring in the same organism. The equation for probability can be found in the Methods section. A long table format of the matrix can be found in *Supplementary file 2*. Additional raw data can also be found in *Supplementary file 1*.

The online version of this article includes the following figure supplement(s) for figure 3:

**Figure supplement 1.** Count of organisms that contain multiple HCP genes.

In this study, we contribute to painting a holistic phylogenetic picture of CODHs by focusing on the analysis of their genetic environment as well as harnessing the concept of synteny in which we use a semi-quantitative approach to predict characteristics of CODH, clade- and subclade-specific. We are presenting certain clade-specific trends in the operon composition in CODH. We limit our search on operon composition because the majority of our data are prokaryotic genomes, which link functional-related genes physically in an operon (*Yaniv, 2011*). We furthermore wanted to limit noise and false positives as much as possible; therefore, we excluded data not within our defined parameters (see Methods). Additionally, since many of the genomes included in this analysis are not completely sequenced or include multiple CODH from different clades, only including genes in close proximity to the target gene ensures that the data is not distorted. Since it is known that many organisms have multiple isoforms of CODHs coded in their genome, we analyze the co-occurrence of CODH of different clades in an organism, as well as the co-occurrences of CODH and HCP. With our findings, we want to propose a systematic approach in the analysis of new CODHs, with the focus on identifying promising $CO_2$ reduction catalysts, suitable for biotechnological application.

## Results

### Co-occurrence and correlation

After evaluating the assemblies in regard to their count of CODH, we found that around 30% of all assemblies encode for more than one CODH. For HCP, this number is much smaller, around 6%. *Figure 3A* shows that the occurrence of multiple isoforms from specific clades within organisms varies. Clades B, C, and D almost exclusively occur only once within a genome, while clades A, E, and F are more likely to co-occur with another isoform from the same clade. The overall trend that we observe is most likely underrepresenting the number of genomes encoding multiple CODHs, since incomplete genomes are also included in these analyses. When calculating the correlation of the co-occurrence of CODH from two different clades in one organism, a pattern evolves

(*Figure 3B*). Most obvious is the low co-occurrence of clades A and E with F. However, clade F reversely has a higher co-occurrence with clade E due to the asymmetrical nature of the data. This is an effect of the different sizes of the clade datasets. Furthermore, all clades seem to have low co-occurrence with clade B. Due to the effect stated above, clade B does, however, show some co-occurrence with clades D and E. Clade C shows strong co-occurrence with clade E, while clade D co-occurs with clades A, E, and F. Clades C and D also have a higher probability to co-occur with each other. As outlined in the Introduction, from biochemical studies, it is known that CODHs from clades A, E, and F are active, whereas $CO_2$/CO interconversion activity is missing in CODH from clade D. This co-occurrence might suggest that the redox sensing properties of CODH from clade D (and potentially clade C) are useful for organisms already containing a functional CODH. Interestingly, a high co-occurrence was also seen for CODH and HCP, with an exception for clade A CODH.

## Neighbor analysis

We semi-quantitatively evaluated the operon composition of 1351 CODHs (121 A, 130 B, 168 C, 253 D, 434 E, 245 F, *Supplementary files 6 and 7*) with proteins whose function we could predict using eggNOG (*Cantalapiedra et al., 2021*; *Huerta-Cepas et al., 2019*), the NCBI product prediction, and manual curation. The results are summarized in *Figure 4*. Even though eight clades of CODH are known, we only present six in this analysis, since our quality measures excluded clade G and clade H CODH from the initial dataset because of poor assembly quality or lack of host information. In the following, we only report on neighbors that are encoded in the same operon as more than 10% of CODHs per clade (see *Supplementary file 2*). Starting with CODHs from clade A, 93% contain a one-carbon pool-related gene in their operon, followed by CooC (62%) and iron-sulfur (FeS) cluster-containing protein (31%). As one carbon pool-related gene, we defined genes associated either with direct conversion of one carbon compounds, such as formate dehydrogenases or with the Wood-Ljungdahl pathway. Clade B CODH operons mainly encode ABC transporter-associated genes (64%). Furthermore, almost a quarter (24%) of all CODHs from clade B could not be associated with any neighbor, and 12% are coded close to transcriptional regulators. For CODH from clade C, the three main neighbors are proteins associated with FeS cluster containing proteins (such as CooF) (72%), NAD(P) or FAD-dependent oxidoreductases (71%), and transcription (58%), or other (10%) regulation. The overall diversity of neighboring proteins from clade D and the fact that a major part of those CODHs seemingly do not encode close to any other genes (64%) made it challenging to sum up their different codons, and no clear pattern could be observed. Only transcription regulation proteins (9.9%) and general regulatory proteins (9.5%) are worth mentioning in this context. Clades E and F both have a larger set of proteins frequently observed in their associated operons. Operons encoding either clade E or F CODH contain CooC-like genes (59% E, 68% F), one carbon pool-associated genes (49% E, 37% F), and FeS genes (29% E, 53% F). Transcription regulators (17% E, 35% F) and NAD(P)/FAD-dependent oxidoreductase (22% E, 42% F) have also been found. The maturation protein CooT was exclusively found in operons from clade E (16%) and F (6.1%). The same holds true for CooJ, but in clade F, CooJ was seen in even fewer operons (12% E, <5.0% F). Additional hydrogenases (25%) and their maturation machinery (17%) are coded primarily for clade F CODH, as well as different types of transporter proteins (11%).

We performed a similar analysis of the operons encoding for HCP and analyzed a total of 1476 HCP genes (class I: 1049, class II: 23, class III: 404, *Supplementary files 8 and 9*) showing a low frequency of isoforms within organisms (*Figure 3—figure supplement 1*). HCP exhibited a large variety of neighbors, leading to difficulties in extracting meaningful information from their operon composition (*Figure 4—figure supplement 1*). Furthermore, classes I and III had a high proportion of entries without any neighbors (49% and 75%, respectively), which is reflected in their tendency to have fewer proteins coded in their operons (see *Figure 4B*). However, we observed a high frequency of FeS cluster proteins (18%) and transcription regulators (17%) for class I, as well as NAD(P) or FAD oxidoreductases (96%) and transport proteins (78%) for class II HCP. It needs to be noted, however, that our sample set for class II HCP is very small, so its information value is considerably lower compared to the other classes/clades.

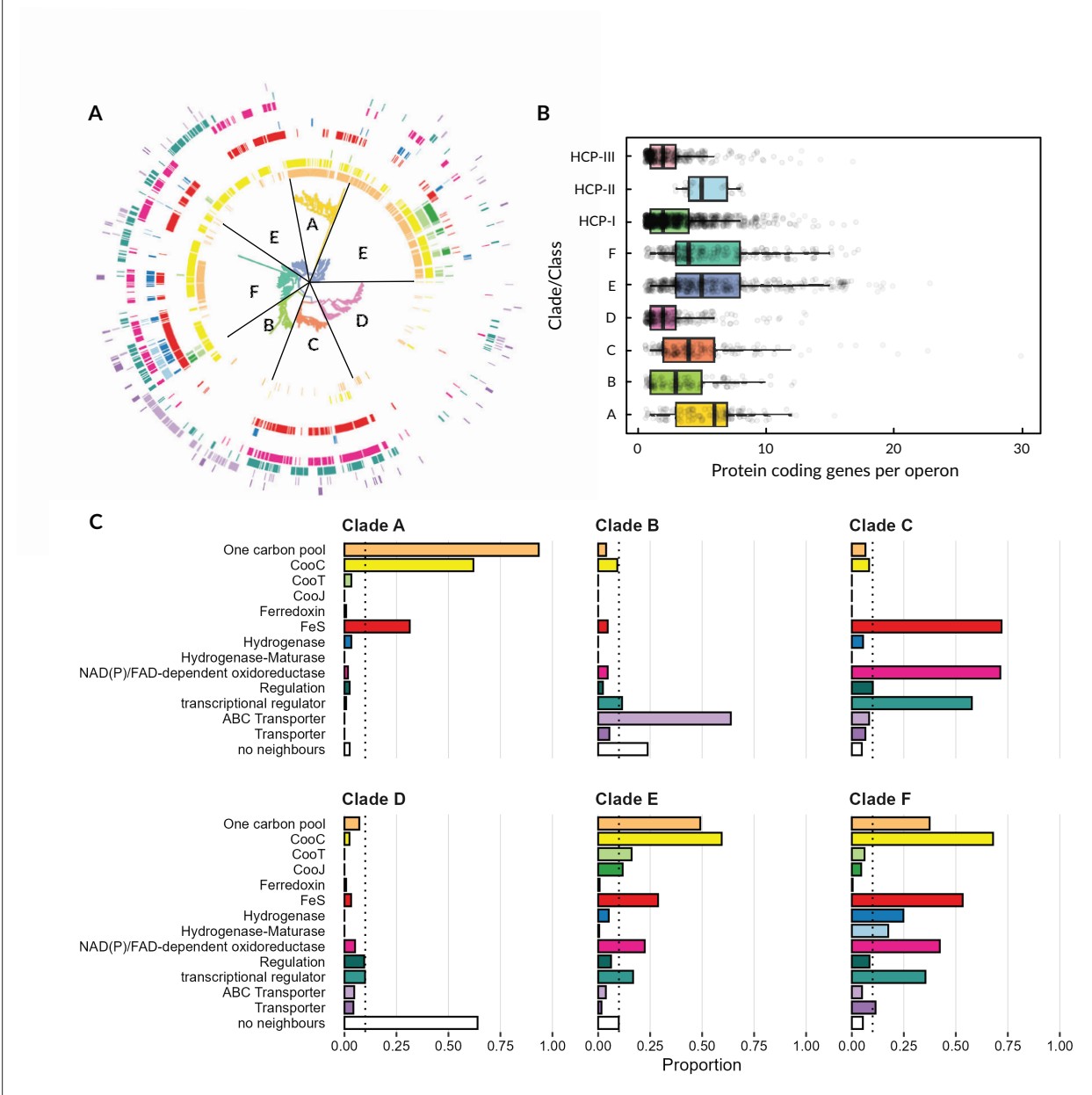

**Figure 4.** Operon content of CODH encoding operons in different clades. (**A**) Phylogenetic tree of putative [NiFe]-CODH unrooted with 1376 sequences. The colored rings mark if the operon from a protein/leaf contains one or more of a certain type of protein using the same coloring as in **C**. The types of proteins are from center to outer ring: one carbon pool, CooC, CooT, CooJ, ferredoxin, FeS, hydrogenase, hydrogenase-maturase, NAD(P)/FAD-dependent oxidoreductase, regulation, ABC transporter, and transporter. A detailed searchable tree with bootstrap values can be found in *Supplementary files 6 and 7*. (**B**) Distribution of operon size for CODH and HCP genes. (**C**) Proportion of [NiFe]-CODH from one clade being coded near a certain type of protein. The dotted line marks a proportion of 10%, which we set as our significance cutoff. Raw data can be found in *Supplementary file 1*.

The online version of this article includes the following figure supplement(s) for figure 4:

**Figure supplement 1.** Selected neighbors of HCP genes and their proportion in regard to all HCP from one subclass.

## Discussion

Our analysis reveals substantial diversity in the occurrence, co-occurrence, and genomic context of CODH and HCP genes, suggesting complex evolutionary and functional relationships within and across microbial lineages. The observed differences in the frequency of multiple isoforms per genome (~30% for CODH versus ~6% for HCP) indicate that CODHs are more often retained in multiple

copies, potentially pointing to functional diversification among its isoforms. Similar values have been reported by *Techtmann et al., 2012*, where they found that a striking 43% of organisms coded for more than one CODH. On the other hand, *Katayama et al., 2024*, investigating only the human gut microbiome, found a number as low as 5.5%. We suspect that this number underrepresents the amount of organisms carrying multiple isoforms of CODH in the human gut, since Katayama and co-workers performed data refinements that exclude potential CODHs (such as the strict requirement for a [4Fe-4S] D-cluster, even though *Inoue et al., 2018* reported on the diversity of the D-cluster earlier; ).

There are many examples of organisms coding for multiple CODH isoforms, as outlined in *Figure 3*; *Supplementary files 10 and 11*. Many of them have been known to literature for a long time (even though their CODH abundance has only been discussed sporadically), with the most famous example being *C. hydrogenoformans* encoding five different CODHs (*Wu et al., 2005*). Another interesting example is *Clostridium formicoaceticum*, since this organism has a total of six CODH isoforms encoded in its genome. It needs to be noted that in our analysis, this organism did not show up as an organism with six CODHs; see *Figure 3A*. This is due to our analysis only counting CODH stemming from the same organism when their genes are associated with the same genome assembly. We therefore rather underestimate counts of organisms with multiple CODHs, such as the aforementioned. The only CODH from *C. formicoaceticum* that has so far been isolated, characterized, and discussed is one of its clade E CODHs, that is associated with acetyl-CoA synthase (ACS) (*Bao et al., 2019*; *Diekert and Thauer, 1978*). Other examples from literature attempted to investigate the influence of CODH isoforms on the metabolism. *Archaeglobus fulgidus* contains three CODH genes, two from clade A and one from clade D. Its clade D CODH seems to have no role in the CO metabolism of this organism (*Hocking et al., 2015*). There also has been a report for the organism *Methanosarcina acetivorans*, which harbors three CODHs, all of them belonging to clade A, where only two of them are associated with ACS and are believed to be involved in the CO metabolism, the other one being a lone gene that is seemingly not involved in carbon metabolism (*Matschiavelli et al., 2012*). Another interesting example is *Thermoanaerobacter kivui*, formerly known as *Acetogenium kivui* (*Collins et al., 1994*). When *Tk*CODH-I (clade C) is deleted from the organism, the strain loses its ability to grow on CO; however, if grown on $H_2+CO_2$ the overall acetate production is greatly increased (*Jain et al., 2022*). Similar effects have been shown for *Clostridium autoethanogenum*, which contains three isoforms of CODH. If its clade C CODH is deleted, the organism's lag phase is reduced and its growth rate is greatly increased (*Liew et al., 2016*). The other two CODH isoforms from this organism are from clades E and D. Deletion of clade D CODH showed no immediate effect on the organism, except moderately lower overall biomass yield (*Liew et al., 2016*). In our analysis, we saw an increased frequency of co-occurrence of clades C and D with A, E, or F, which together with biological data may indicate a complementary role, possibly linked to redox sensing or regulatory functions. This is especially evident with the examples for clade C CODHs from *T. kivui* and *C. autoethanogenum*. For the case of clade D, which lacks catalytic activity toward $CO/CO_2$ interconversion (as reported by *Jeoung et al., 2022*, through their recombinant production of *Ch*CODH-V) and until now has unknown influence in the metabolism that might only manifest in harsher environments, since it's believed to be involved in stress response (*Jeoung et al., 2022*) (similar to HCPs [*Hagen, 2022*], see below). However, experimental proof for this claim is still missing.

Furthermore, clades A, E, and F rarely co-occur. Interestingly, many organisms do, however, contain multiple copies of CODHs from one of these clades, such as *M. acetivorans* (clade A), *C. hydrogenofromans* (clade F), and *C. formicoaceticum* (clade E). We suspect an evolutionary reason behind this, as is also outlined by others (*Adam et al., 2018*; *Lindahl and Chang, 2001*). Biochemical data indicate that CODHs from these clades possess $CO/CO_2$ interconversion capability, as previously mentioned. This is also in line with the genetic context of these CODHs, which is most often tuned for this $CO/CO_2$ interconversion chemistry (*Figure 4C*, see below).

The high rate of co-occurrence between CODH and HCP genes (except for clade A) suggests functional integration, a shared metabolic niche, or involvement in a coordinated response to redox stress, given that HCPs are thought to regulate nitric or oxidative stress (*Hagen, 2022*). The lack of co-localization for clade A CODHs might point to distinct metabolic roles or evolutionary constraints, or to the high rate of archaeal genes in clade A CODHs, even though HCP genes are known to also be found in archaea (*Hagen, 2022*). Interestingly, the co-occurrence between clade D CODH and

HCP seems the highest for our dataset; the reason for that remains unclear. Similar to clade D CODH, empirical proof that HCP influences the activity or expression of CODH is missing.

The genomic context analysis adds another layer of functional inference, as has been done before by others with different foci (*Inoue et al., 2018*; *Katayama et al., 2024*; *Matson et al., 2011*; *Techtmann et al., 2012*). We could show again that operons containing clade A CODHs are highly conserved with one carbon pool-related genes and CooC, as they are almost exclusively found as part of the Wood-Ljungdahl pathway, which has a typical arrangement similar to *Methanosarcina barkeri* CODH (*Mb*CODH, *Figure 2*, *Supplementary file 7*). Recently, another representative from this group from *M. thermophila* (*Met*CODH) has been resolved (*Biester et al., 2024*).

In contrast, clade B CODHs appear largely alone or associated with transport-related genes, raising the possibility of a non-canonical or even degenerated function. Its operon composition is also rather consistent, and its arrangements only vary to a small extent, as ABC transporters are either coded upstream (as for *Ruminococcus flavefaciens*' CODH, *Rf*CODH, see *Figure 2*) or downstream of the CODH gene. Almost all operons analyzed do not contain any maturases, except for a small cluster that branches off rather early in the tree (*Supplementary file 7*). This might indicate that the need for a maturase was lost due to re-purposing of the CODH. We yet await biochemical characterization of any clade B CODH.

Clade C CODHs are associated with FeS cluster proteins, regulators, and redox enzymes, pointing toward more regulatory or redox-modulatory roles, which could also be indicated in knockout studies (*Liew et al., 2016*). The only isolated example from this clade is *Tk*CODH-I. Its operon exhibits a composition only partially representable for clade C CODHs, containing only one other gene coding for an FeS protein (*Figure 2*). Furthermore, *Tk*CODH-I's sequence branches off early and seems to be rather distinct (*Supplementary file 5*), only having one other close relative from *Aceticella autotrophica* (*Frolov et al., 2023*). Furthermore, the reported isolated CODH from *Jain et al., 2022*, stems from a CO-adapted strain (*Weghoff and Müller, 2016*), which might harbor mutations in the protein sequences that are not accessible to us at the moment. Drawn together, we conclude that right now, *Tk*CODH-I might not be an optimal representative for clade C CODHs, and more clade C CODHs should be isolated to help us understand their biochemical properties better.

The high operonic variability and frequency of solitary coding regions in clade D might reflect either evolutionary drift or multifunctionality not restricted to operonic structure. Clade D will therefore not be discussed further.

Operons from clades E and F are more functionally complex, including components from the Wood-Ljungdahl pathway, hydrogenases, and additional redox partners, consistent with a diverse metabolic role. That being said, their operon compositions and arrangements showcased some interesting clustering. Starting with clade F, which has the highest proportion of CODHs that might be associated with hydrogenases, including the aforementioned *Ch*CODH-I and *Rr*CODH, both interacting with hydrogenases to produce hydrogen in vivo (*Fox et al., 1996*; *Soboh et al., 2002*), however with greatly differing operon compositions (*Figure 2*). *Ch*CODH-I-like operons (see *Supplementary file 7*) contain their hydrogenase modules directly within the CODH operon, whereas *Rr*CODH-like operons do not include the hydrogenase module which is coded upstream of the CODH gene, with an intergenic space of >400 bp (*Fox et al., 1996*). *Rr*CODH-like operons are the only clade F operons that include two additional maturation enzymes, CooT and CooJ. Clade F also contains many ACS-associated CODHs, such as *Neomoorella thermoacetica* CODH, *Nt*CODH, formerly known as *Moorella thermoacetica* (*Gtari and Ventura, 2025*). Those *Nt*CODH-like operons all have the same arrangement. This arrangement is distinct from clade A and E ACS-associated CODH operons.

In our dataset, most of the hydrogenases and their maturation genes are associated with clade F, suggesting active hydrogen metabolism, coupling CO oxidation to $H_2$ production or consumption, as has been suggested earlier for a wider range of CODH clades (*Inoue et al., 2018*; *Techtmann et al., 2012*). We believe that our data grossly underestimates this relationship overall, since operon examples such as *Rr*CODH and *Tc*CODH-II showcase that hydrogenases associated with a CODH are not necessarily encoded in the same operon. There has also been a report of a clade A CODH from the methanogen *M. thermophila* (*Terlesky and Ferry, 1988*) being associated with a hydrogenase. However, in later studies, it was shown that the genome of *M. thermophila* does not contain a hydrogenase (*Smith and Ingram-Smith, 2007*). Investigation of the electron transport chain of its membrane could not find a hydrogen-oxidizing complex (*Welte and Deppenmeier, 2011*). Together

with our analysis, we conclude that hydrogenase association is a trait almost exclusive to clade E and F CODHs. ChCODH-II's operon seems to be rather uniquely constructed, as a similar operon composition only can be found for other *Carboxydothermus* species. A similar situation can be seen for *Ch*CODH-IV, where its operon containing FeS- and NAD/FAD-dependent oxidoreductases is closer in similarity to some clade E CODH.

Regarding the biggest clade, clade E, its diversity is striking. *Nv*CODH, formerly known as *Dv*CODH (*Waite et al., 2020*), from the organism *N. vulgaris*, has a very small operon with only two genes in its close proximity, a transcriptional regulator (*Zhou et al., 2012*) and a maturation enzyme (CooC), see *Figure 2*. This is seen for a huge number of both clade E and F CODHs. The occurrence of neither CooJ nor CooT is prominent, both only appearing in two very distinct parts of clade E, all of them being associated with one carbon pool metabolism, with one exception from *Clostridium pasteurianum* BC1. This operon resembles the clade F *Rr*CODH-like operon. The previously introduced archaeal CODHs, *Tc*CODH-I and *Tc*CODH-II from clade E, are contained in operons (*Figure 2*) that are rather specific and only found for a few other *Thermococcus* or *Pyrococcus* species (*Benvenuti et al., 2020*; *Kim et al., 2015*). It needs to be noted that *Tc*CODH-I's CooC gene is coded outside of its operon, and on the opposite strand. The CooC gene is therefore not included in our analysis. We are only aware of examples from this type of operon and don't expect this to be a common trait of the CODH maturation machinery. However, it needs to be noted that the CooC proportion might be slightly underestimated. Interestingly, *Tc*CODH-II, like CODH, all contain a CooT-like protein in their operon, forming the only cluster of CODH that contains only CooT-like proteins without CooJ. Within clade E, another unique genomic neighborhood from *Tk*CODH-II must be pointed out. From experimental data, it is known that this CODH is associated with ACS (*Jain et al., 2022*); however, in our analysis, we did not see this ACS complex in *Tk*CODH-II's operon. This is due to the ACS subunit being coded further downstream of the CODH gene, not being taken into account due to our initial parameters, which showcase the limits to this study.

For HCPs, the high variability and low operon density – especially in classes I and III – point toward more modular or conditionally expressed roles, similar to clade D CODH. The clear patterning in class II operons, though based on a limited sample, may reflect specialized functions, perhaps in niche-specific oxidoreductase activities.

## Conclusion

As previously mentioned, the aim of this study is to identify which CODH clades harbor the most promising enzymes for future application in $CO_2$ reduction. The operon composition of CODHs from different clades shows distinct differences, and what we could gather from this information is that clades A, E, and F are the most likely clades to harbor CODHs able to efficiently convert $CO_2$ to CO. These clades are therefore the most interesting for $CO_2$ reducing biotechnological applications, or as inspiration for new synthetic catalysts. Also, literature has shown that the activity of many CODHs depends on co-expression with maturation proteins such as CooC. In some cases, CooJ and CooT are also required for full activation. Although some CODHs (most notably CODH-II from *C. hydrogenoformans*) can function independently of maturases, our neighborhood analysis indicates that maturase-coding genes are predominantly found in operons from clades A, E, and F. This pattern implies yet again that these clades may represent more biochemically active or catalytically optimized CODHs, making them promising targets for future functional studies and biotechnological applications.

The function of clade B could not be deduced based on its genomic environment, but it seems to have a remarkable self-standing function that is not shared with any other CODHs. Its low co-occurrence with other CODH clades within organisms also supports this unique role for clade B. Clades C and D are more likely to show low or even no activity toward $CO_2$/CO interconversion, as was deduced from literature and the lack of one carbon metabolism-related genes in their operons. However, Jain et al. recently showed low CO oxidation activity in a clade C CODH from *T. kivui*, but this enzyme originated from a strain that had acquired the ability to grow on CO through laboratory evolution (*Jain et al., 2022*). Sequence data used to classify the CODH into clade C was from the original strain (incapable of growing on CO), and data on the engineered strain is not available. It is therefore not known whether the active CODH is the wild-type or an engineered enzyme, and we cannot draw any conclusions regarding the activity of clade C CODHs. Taken together, this makes clades B, C, and D

less promising in the hunt for $CO_2$ reduction catalysts. However, much is still unknown about these enzymes, such as their cellular function.

Future work should focus on experimentally validating the functional differences among CODH isoforms, especially in organisms that contain members of multiple clades. Caution is warranted when extrapolating enzymatic activity or inactivity from a limited number of characterized examples to entire clades. Additionally, transcriptomic and proteomic studies could illuminate condition-dependent expression patterns and confirm proposed regulatory functions. Future bioinformatic work should look at the co-occurrence of CODH with other proteins outside their operons, once more sequence data is available. Finally, deeper phylogenomic analyses may reveal the evolutionary drivers behind the observed distribution and diversification of these ancient redox enzymes.

# Methods
## Data collection and refinement

Multiple pBLAST (*Madden, 2013*) searches (BLOSUM62, E<0.05, blastp 2.16.1+, NCBI online web server, nr database accessed 2025-01-16) were carried out using NCBI accession numbers provided by *Inoue et al., 2018* (A-1, WP_011305243; A-2, WP_010878596; A-3, OGW06734; A-4, OIP92259; A-5, ODS42986; A-6, OIP30420; B-1, WP_026514536; B-2, WP_015485077; B-3,WP_012645460; B-4, WP_011393470; C-1, WP_039226206; C-2, WP_013237576; C-3, WP_010870233; C-4, WP_044921150; D-1, WP_011342982; D-2, WP_015926279; D-3, WP_079933214; D-4, WP_096205957; E-1, WP_012571978; E-2, WP_010939375; E-3, WP_088535808; F-1, WP_011343033; F-2, WP_011389181; G-1, OGP75751) and (*Techtmann et al., 2012*) (mini CooS, WP_007288589.1). Accession numbers from CODH from clade H (*Inoue et al., 2022*) were initially omitted due to limited host information. Duplicates were removed using seqkit's (*Shen et al., 2016*) rmdup v2.9.0. Sequences of length below 400 amino acids were removed. Clustering was performed to further reduce data size, by using cd-hit v4.8.1 (*Li et al., 2002*; *Li et al., 2001*; *Li and Godzik, 2006*) and a global sequence identity of 99% or 90%, the latter only used for tree generation. It was necessary to have high sequence similarity in the clustering within organisms, since it was known that some organisms have multiple CODH with striking sequence similarities in their genome such as *Clostridium pasteurianum* BC1 (taxid: 86416) that contains WP_015614757.1 and WP_015615315.1 with 93.27% similarity. For the dataset involved in neighbor analysis, taxonomic information for each sequence was retrieved using R-packages taxize v0.10.0 (*Chamberlain et al., 2020*; *Chamberlain and Szöcs, 2013*) and taxizedb v0.3.2 (*Chamberlain et al., 2025*), and only sequences that could be related to a recorded organism were kept (*Supplementary file 3* and *Supplementary file 4*). Sequences were aligned using E-INS-I from mafft v7.526 (*Katoh and Standley, 2013*) and sequences that had gaps in important positions related to D, B, or C cluster or acid-base active site residues were sorted out. The alignment was trimmed using trimAl's (*Capella-Gutiérrez et al., 2009*) automated1 option v1.4.rev22 and a tree was generated using Fast-Tree v2.2 (*Price et al., 2010*). Via visual inspection, further sequences that were not CODH sequences were removed. Leaves from unusually long branches were examined manually. If a sequence from such a branch was annotated as a protein other than hydroxylamine reductase or carbon monoxide dehydrogenase (CODH), it was investigated further. We assessed whether the protein length exceeded 400 amino acids and whether the key clusters were present. If these criteria still yielded ambiguous results, the protein structure was predicted using AlphaFold 3 (*Abramson et al., 2024*) to determine whether it adopted the characteristic CODH fold. Sequences lacking this fold were discarded. The final list of CODH sequences used in the neighbor and correlation analysis counted 1376. A similar approach was done for HCP (class I, Q01770.2; class II, WP_000458809.1; class III, WP_013294878.1), and a final count of 1545 sequences was collected for neighbor and correlation analysis. We applied the same procedure described for the previous dataset to a second set of CODH genes curated at 90% cd-hit identity, yielding 5508 sequences. See *Appendix 1—figure 1* for detailed flowchart. Custom code can be found and retrieved for GitHub (https://github.com/boehmax; *Böhm, 2025a*; *Böhm, 2025b*; *Böhm, 2025c*; *Böhm, 2025d*).

## Neighbor analysis

Genome information was downloaded for the accession lists generated for CODH and HCP. Therewith, 955 and 1425 genomes were downloaded, respectively, from NCBI's genome database using

NCBI-datasets command line tools v18.5.0 (*O'Leary et al., 2024*). Neighboring genes were defined as those located within a maximum of 15 genes upstream or downstream of the target gene, with an intergenic distance not exceeding 300 base pairs (bp), as was done previously by *Inoue et al., 2018*. We decided to use this relatively large intergenic distance to include as many neighbors as possible, and we expect that unrelated genes will disappear in the noise. For the same reason, we included an overlap region of 50 bp for genes in the same operon, which is rather high, as genes, for example, in *Escherichia coli* usually overlap 1–4 bp (*Johnson and Chisholm, 2004*). Amino acid sequences for those genes were retrieved from the NCBI nr protein database using Entrez v23.5 (*Sayers, 2022*). Their function was predicted using eggNOG v5.0 (*Huerta-Cepas et al., 2019*) and eggNOG-mapper v2.1.12 (*Cantalapiedra et al., 2021*). We considered results from eggNOG, as well as product predictions from NCBI, when manually assigning selected functional groups. The data was plotted using R v4.4.3 (*R Development Core Team, 2023*), tidyverse v2.0.0 (*Wickham et al., 2019*), patchwork v1.3.1 (*Pedersen, 2025*), ggnewscale v0.5.2 (*Campitelli et al., 2025*), ggtree v3.14.0 (*Yu et al., 2018*; *Yu et al., 2017*), ggtreeExtra v1.16.0 (*Xu et al., 2021*), treeio v1.30.0 (*Wang et al., 2020*), and gggenes v0.5.1 (*Wilkins, 2023*). Since CooJ determination was neither possible with the NCBI prediction nor via eggNOG, we selected operons from clades E and F that contained CooS and CooT and manually extracted accession numbers of potential CooJs which were used to search for further accession numbers using PSI-BLAST (BLOSUM45, E<0.001, NCBI online web server, nr database accessed 2025-04-16). The summary can be found in *Supplementary file 1*. These accessions were used to help annotate potential CooJs in our analysis. We could identify 68 potential CooJ genes.

## Correlation analysis

Correlation coefficients of CODH and HCP from different clades/classes were calculated according to the formula

$$P\left(X|Y\right) = \frac{N_{XY}}{N_Y},$$

where $N_Y$ is the total number of assemblies containing protein from clade/class Y, $N_{XY}$ is the total number of assemblies containing proteins from both clade/class X and Y, and P(X|Y) is the probability that a genome coding for a protein from clade/class Y also codes for a protein from clade/class X.

## Tree generation

In total, five trees were generated. Trees carrying phylogenetic information were generated via IQ-TREE v2.0.7 (*Minh et al., 2020*) with the LG+I+R10 model and ultrafast bootstrapping with 1000 resampling for a dataset of 5508 CODH sequences, a dataset of 1351 CODH sequences, and a dataset of 1476 HCP sequences (see above for details on their generation). For the 5508 sequence CODH dataset, an outgroup was introduced to root the tree (MBE6442607.1). Sequences were aligned within their dataset using mafft's FFT-NS-2 v7.526. The alignment was again trimmed using trimAl v1.4.rev22 and built using IQ-TREE v2.0.7 with the above parameters. For tree inspection and plotting, ggtree v3.14.0 (*Yu et al., 2017*) was used. The two other trees generated are taxonomic trees, either only on taxid using a custom Python script and ete3 v3.1.3 (*Huerta-Cepas et al., 2016*), or from WoL: Reference Phylogeny for Microbes (*Zhu, 2023*; *Zhu et al., 2019*). Clades of CODH were defined as described previously by others (*Inoue et al., 2022*; *Inoue et al., 2018*; *Techtmann et al., 2012*). Our tree showed similar topology with bootstraps of >75% for all clades.

## Acknowledgements

The Novo Nordisk Foundation (Grant reference number NNF21OC0066716) is gratefully acknowledged for funding.

## Additional information

### Funding

| Funder | Grant reference number | Author |
|---|---|---|
| Novo Nordisk Fonden | NNF21OC0066716 | Maximilian Böhm<br>Henrik Land |

The funders had no role in study design, data collection and interpretation, or the decision to submit the work for publication.

### Author contributions

Maximilian Böhm, Conceptualization, Investigation, Methodology, Writing – original draft, Writing – review and editing; Henrik Land, Conceptualization, Supervision, Funding acquisition, Project administration, Writing – review and editing

### Author ORCIDs

Maximilian Böhm ⬛ https://orcid.org/0000-0003-0205-8030
Henrik Land ⬛ https://orcid.org/0000-0003-3073-5641

Reviewer #1 (Public review): https://doi.org/10.7554/eLife.108780.3.sa1
Reviewer #2 (Public review): https://doi.org/10.7554/eLife.108780.3.sa2
Author response https://doi.org/10.7554/eLife.108780.3.sa3

## Additional files

### Supplementary files

Supplementary file 1. Contains information regarding neighbor analysis.

Supplementary file 2. Contains data on the co-occurrence of CODH/HCP in one organism. This data was used to create heatmaps in *Supplementary files 10 and 11*.

Supplementary file 3. Contains initial sorting of CODH accession numbers based on their source. Only sources that were classified as organisms passed and were further used for neighbor analysis.

Supplementary file 4. Contains initial sorting of HCP accession numbers based on their source. Only sources that were classified as organisms passed and were further used for neighbor analysis.

Supplementary file 5. CODH phylogenetic tree with 5808 leaves. Nodes supported with bootstrap values above 0.90 are marked with an orange dot. Ultrafast bootstrapping was performed using 1000 resamplings. Rooted using Desulfovibrio desulfuricans' HCP (MBE6442607.1).

Supplementary file 6. CODH phylogenetic tree with 1376 leaves. Nodes supported with bootstrap values above 0.90 are marked with an orange dot. Ultrafast bootstrapping was performed using 1000 resamplings. Unrooted.

Supplementary file 7. Same tree as Tree1 with additional operon information.

Supplementary file 8. HCP phylogenetic tree with 1476 leaves. Nodes supported with bootstrap values above 0.90 are marked with an orange dot. Ultrafast bootstrapping was performed using 1000 resamplings. Unrooted.

Supplementary file 9. Same tree as Tree3 with additional operon information.

Supplementary file 10. Tree based on taxid using ete3 (*Huerta-Cepas et al., 2016*) from organism with heatmap indicating occurrence of CODH or HCP in an organism. For detailed values, please refer to *Supplementary file 2*.

Supplementary file 11. Tree based on tree generated by WoL (*Zhu, 2023*) from organism with heatmap indicating occurrence of CODH or HCP in an organism. For detailed values, please refer to *Supplementary file 2*. This tree contains fewer organisms than Tree5, because of limited organisms found in WoL's dataset (*Zhu, 2023*; *Zhu et al., 2019*).

MDAR checklist

## Data availability

All codes for bioinformatic analysis presented in this paper are openly accessible at Zenodo under the following DOI's; https://doi.org/10.5281/zenodo.16736767, https://doi.org/10.5281/zenodo.16736754, https://doi.org/10.5281/zenodo.16736722, https://doi.org/10.5281/zenodo.16744414.

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

## Appendix 1

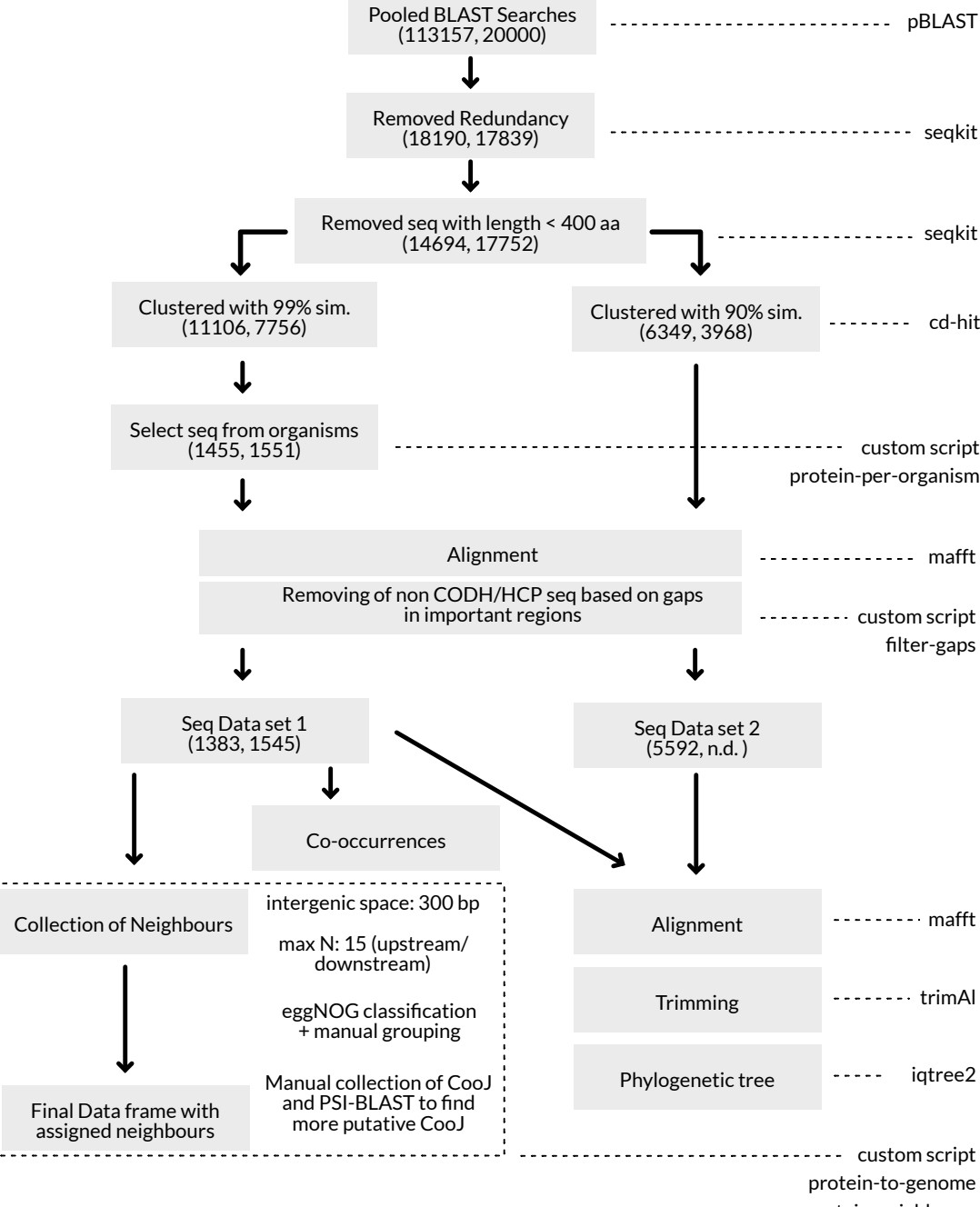

**Appendix 1—figure 1.** Flowchart for the data collection and refinement process. Numbers in braces refer to the number of sequences in each step, the first number referring to putative CODH sequences, the second number refers to putative HCP sequences. Custom scripts are available online (see ***Böhm, 2025a***; ***Böhm, 2025b***; ***Böhm, 2025c***; ***Böhm, 2025d***).

