## [Editor Report · eLife Assessment]

This **valuable** work analyzes a large dataset of [NiFe]-CODHs, integrating genomic context, operon organization, and clade-specific gene neighborhoods to discern patterns of diversification and adaptation. A consistent examination of CODH genomic contexts, including CODH-HCP co-occurrence, informs interpretations of enzymatic activity, biotechnological potential, and differential functional roles, in line with current standards in genomic enzymology. With **solid** support, this work provides a broadly informative contribution to the field.

---

## [Referee Report · Reviewer #1 (Public review)]

Summary:

This manuscript analyzes a large dataset of [NiFe]-CODHs with a focus on genomic context and operon organization. Beyond earlier phylogenetic and biochemical studies, it addresses CODH-HCP co-occurrence, clade-specific gene neighborhoods, and operon-level variation, offering new perspectives on functional diversification and adaptation.

Strengths:

The study has a valuable approach.

Comments on revised version:

I am satisfied that the authors have adequately addressed my previous comments in the revised manuscript.

---

## [Referee Report · Reviewer #2 (Public review)]

The authors present a comparative genomic and phylogenetic analysis aimed at elucidating the functions of nickel-dependent carbon monoxide dehydrogenases (Ni-CODHs) and hybrid-cluster proteins (HCPs). By examining gene neighborhoods, phylogenetic relationships, and co-occurrence patterns, they propose functional hypotheses for different CODH clades and highlight those with the greatest potential for biotechnological applications.

A major strength of this work lies in its systematic and conceptually clear approach, which provides a rapid and low-cost framework for predicting the functional potential of newly identified CODHs based on sequence data and genomic context. The analysis is careful in minimizing false positives and offers valuable insights into the diversity and distribution of CODH enzyme clades.

---

## [Author Response]

The following is the authors’ response to the original reviews.

**Public Reviews:**

**Reviewer #1 (Public review):**
(1) Rationale for excluding clades G and H and clarification of clade definitions

We appreciate this important request for clarification. In the revised manuscript, we now explicitly state (Methods, Tree generation) that the phylogenetic framework used in this study follows the clade definitions established by Techtmann et al. (Front. Microbiol. 2012, 3, 132), which classify [NiFe]-CODHs into clades based on high supporting values in nodes (bootstrap >75). We deem Techtmann et al.’s work as best lead, since their approach with two different types of trees (ML vs. Bayesian) gives solid support to this classification of clades. We ourselves did not perform Bayesian statistics, instead we used the known clades from literature to assign ours.

Clades G and H were not deliberately excluded from downstream genomic-context and operon analyses. They were excluded by our pipeline, because their data did not fulfil our initial quality assessments, such as: host classified down to species level (https://github.com/boehmax/protein-per-organism), and protein exists in the IPG database of NCBI (https://github.com/boehmax/protein-to-genome).

Clade G and H are both represented by only a very small number of sequences, most of which derive from fragmented or poorly annotated genomes, preventing reliable assessment of operon organization and gene neighbourhood conservation. As a result, inclusion of these clades would not allow statistically meaningful or biologically interpretable comparisons with clades A–F.

To improve transparency, we have added a brief explanation of these limitations in the Results (Results, Neighbor analysis).

(2) Presentation and interpretation of co-occurrence data

We agree that the presentation of the co-occurrence data required improvement. In the revised supplementary material, we now include a table in the long format that might be easier to interpret than a matrix representation as seen in Fig. 3B.

We have also revised the Results text to more precisely reflect the numerical trends. Specifically, we clarify that clade D shows co-occurrence with clades A, E and F, while clade C only displays co-occurrence with clade E. The statement that clades C and D “more often co-occur” has been removed and rephrased to avoid overgeneralization and to better align with the quantitative data shown in Figure 3B and the supplementary table (Results, Co-occurrence and Correlation).

(3) Rationale for operon-level rather than organism-level analysis

We thank the reviewer for highlighting this conceptual point. In the revised manuscript, we now explicitly state that our analysis was conducted at the operon level because individual genomes frequently encode multiple CODH operons that are phylogenetically and functionally distinct. Treating each operon as an independent functional unit allows us to capture this intra-genomic diversity and to associate specific gene neighbourhoods with individual CODH clades. We furthermore discuss in the introduction explicitly technical reasons why we decided to limit this study to the operon level for more transparency.

Nevertheless, we acknowledge that this approach may overlook higher-level regulatory or physiological interactions among multiple CODHs encoded within the same genome. This limitation is now discussed explicitly, and we acknowledge that operon-level analysis should be a complementary, not exhaustive, framework for functional inference.

**Reviewer #2 (Public review):**

We thank Reviewer #2 for their positive assessment of the conceptual clarity and methodological utility of our approach, as well as for their thoughtful discussion of its limitations.

Regarding incomplete genome assemblies, limited representation of class II HCPs, and potential omission of distal pathway components, we agree fully. We stress that our conclusions are probabilistic and hypothesis-generating rather than definitive functional assignments.

In response to the concern about reproducibility of the visual filtering step, we have added a more explicit description (Methods, Data collection and refinement) of the criteria used to exclude non-CODH homologs (e.g., absence of conserved active-site motifs, unknown folds predicted with AlphaFold3, extremely long tree branches). This clarification improves transparency and facilitates replication of the analysis.

Finally, we concur that extrapolating enzymatic activity or inactivity from a limited number of characterized representatives should be done cautiously. We have revised the wording throughout the manuscript to further temper such generalizations and to frame our interpretations explicitly as predictions that require experimental validation.

Once again, we thank both reviewers for their constructive feedback, which has significantly improved the clarity, rigor, and transparency of the manuscript. We believe that the revisions address all concerns raised and strengthen the overall contribution of this work.

**Recommendation from authors:**

**Reviewer #1 (Recommendations for the authors):**

All suggested editorial and stylistic corrections were implemented. These include refinements to the wording in the Abstract, grammatical corrections, streamlined phrasing, standardized figure callouts and supplementary file references, corrected abbreviations, and consistent formatting of references and author names. The only exception concerns the suggested change from *Met*CODH to *Mt*CODH. We have retained *Met*CODH, as this abbreviation is well established in the literature for the *Methanothermobacter thermophila* CODH and is commonly used in prior studies (e.g., https://doi.org/10.1073/pnas.2410995121). *Mt*CODH has historically been referring to CODH from *Neomoorella thermoacetica* (previously *Moorella thermoacetica*, hence the abbreviation *Mt*). We chose to rename that to *Nt*CODH but to avoid confusion, keep *Met*CODH for *Methanothermobacter thermophila*.

**Reviewer #2 (Recommendations for the authors):**

We likewise addressed the majority of recommendations. We now report the versions of all software tools and databases used, standardized capitalization and naming of software and platforms (e.g., GitHub, eggNOG), clarified the BLAST implementation and database employed, and added direct repository links for custom scripts in both the Methods section and the bibliography. Overall grammatical consistency and formatting were improved throughout the manuscript. In addition, the criteria and procedure used for visual inspection to remove non-CODH sequences are now described more explicitly to enhance reproducibility, and several methodological sections were streamlined as suggested. Minor textual redundancies were removed, and phrasing was simplified where appropriate.

Figure legends and formatting were revised to improve clarity and consistency. Adjustments to color usage and font consistency were made where feasible to enhance readability. The color scheme in Figure 1 was adjusted as suggested, and darker shades were chosen for clade H and G. This change was also implemented in the Supplementary File 9_Tree5. Figure 3A was retained, as it provides information on the frequency of multiple CODHs from the same clade within genomes, which cannot be inferred from the probability matrix shown in Figure 3B; together, these panels offer complementary insights. We adjusted the figure caption to make this clearer. We increased the visibility of data points in Figure 4B. To allow inclusion of the full dataset we did not collapse the x-axis as suggested. Figure 4C was retained in its original format to emphasize the characteristic operon “fingerprints” of each CODH clade, which is a central focus of this work. A table is supplied in Supplementary File 2, which allows data exploration with the preferred focus of the reader.

A small number of suggestions were therefore not implemented exactly as proposed, primarily where alternative revisions were judged to better preserve clarity or analytical intent. These decisions are minor and do not affect the conclusions or reproducibility of the study.

Overall, we believe that these revisions have substantially improved the manuscript’s readability, transparency, and technical rigor, and we thank the reviewers again for their careful and constructive feedback.